# Spatial spillover and impacting factors of green development: A study based on China's provincial data

**Jie Huang[1], Juan Chen[2]\***

1 Business School, Xinyang Normal University, Xinyang, Henan, China, 2 School of Business Administration, Anhui University of Finance and Economics, Bengbu, Anhui, China

\* lynn2012@163.com

## Abstract

Green development, an essential part of sustainable development transformation, is spatially correlated intra- and inter-regionally. However, previous research has not fully addressed the spatial characteristics of green development. This study investigates the spatial correlation structures, core–peripheral positions, and factors impacting the spatial network formation of China's green development. Based on the green development evaluation index system modified by the entropy method, this study applies social network analysis, block model analysis, and quadratic assignment procedure analysis to data from 30 provinces in China. The results confirm the spatial spillover effect is overwhelmingly present in China's green development. The findings further distinguish the core roles of provinces including Hunan, Tianjin, Zhejiang, Henan, and Xinjiang, and underline factors of green economic growth, governmental policy support, spatial adjacency, and geographic distance as significantly affecting the spatial network formation of China's green development. Policy recommendations for green development are then put forward.

## Introduction

China has undergone remarkable economic achievements, benefiting from the country's reform and opening-up policies, but at the cost of substantial energy consumption and extensive environmental pollution. Data show that from 1978 to 2020, China's GDP (Gross Domestic Product) rose from 149 billion to 15.59 trillion US dollars, with an average growth rate of 11.66%, making it the world's second-largest economy. However, China's energy consumption in the year 2020 reached 4.64 billion tons (standard coal), accounting for 23.2% of the world's total energy consumption. China's $SO_2$ and $NO_X$ emissions have reached 5.16 and 12.88 million tons, respectively.

During these decades, energetic and environmental issues have drawn widespread attention in China, both practically and academically. The Chinese State Council issued its Air Pollution Prevention Action Plan in 2013, with a focus on how to control the inhalable particulate matter in the air [1]. In 2018, the 19th National Congress of the Communist Party of China advanced the concept of "green development," stating that China would develop a green economy and expand its investment in environmental protection sectors. China's 14th Five-Year Plan for

**Data Availability Statement:** All data are collected from China Statistical Yearbooks and China's provincial statistical yearbooks from 2006-to 2019. Available at: https://kns.cnki.net/kns/brief/result.aspx?dbprefix=CYFD.

**Funding:** Major Research Projects in Philosophy and Social Sciences of the Ministry of Education of China (grant no.17JZD013, recipient Jie Huang), Support Plan for Scientific and Technological Innovation Talents in Henan Institutions of Higher Learning (humanities and social sciences, grant no. 2021-CX-54, recipient Jie Huang), and Youth Project of National Natural Science Foundation of China (grant no. 72203197, recipient Jie Huang), Nanhu Scholars Program for Young Scholars of Xinyang Normal University (grant no. 2023019, recipient Jie Huang). The funders had no role in study design, data collection and analysis, decision to publish, or preparation of the manuscript.

**Competing interests:** The authors have declared that no competing interests exist.

National Economic and Social Development has proposed several green development goals. By 2025, China's carbon emission intensity should be cut by 18% and energy consumption intensity reduced by 13.5%; the use of non-fossil energy should be increased to 20% of the primary energy consumption; and the number of days with good air quality should account for 87.5% of the entire year.

Besides these crucial green development practices in China, recent literature has also addressed the topic of green development more broadly, from establishing evaluation systems of green development [2–7], exploring spatial heterogeneity and the correlation of regional green development status [2, 8–10], and identifying the influencing factors toward that heterogeneity and correlation [11–17]. Although this literature has laid a sound research foundation, gaps remain. First, prior studies are limited to addressing the spatial correlations within geographically adjacent regions based on a green development that is less than comprehensive. Few studies have measured the overall green development status covering most of the territory of China. More importantly, despite confirmation of the existence of spatial correlations in China's green development, extant research has rarely addressed the structural characteristics of the spatially correlated network and has not identified a holistic set of factors impacting that network.

The present study aims to fill these research gaps by investigating the structural characteristics of the spatially correlated network and impacting factors for China's green development. We first develop an evaluation index system through the entropy weight method and then apply social network analysis (SNA), block model analysis (BMA), and the quadratic assignment procedure (QAP) to China's provincial data from 2006 to 2019.

This study contributes to a comprehensive understanding of the spatial characteristics and impacting factors of China's green development. Specifically, we expand the identification of spatial spillover from within geographically adjacent regions to a broader network constructed by all adjacent and nonadjacent areas. We investigate the characteristics and ascertain each province's core–periphery role in the spatial spillover network. We also highlight certain factors affecting the construction of that network. Our findings have implications for policymakers in promoting green development in China.

## Literature review

Recent research on regional green development can be divided into several aspects around development heterogeneity. First, researchers have focused on establishing an evaluation system of green development levels and efficiencies. Some researchers have developed operational measurements of regional green development level based on the analytic hierarchy process (AHP) entropy method [2, 3], the three-dimensional AHP method [4], the entropy technique for order preference by similarity to ideal solution model in combination with the Theil index method [18, 19], the multi-hierarchy linear summation method [20], or using a software tool to assess energies in multi-scale life cycles [21]. Other researchers have focused on establishing the green total factor productivity to depict the green development efficiency among different regions. For instance, researchers have applied data envelopment analysis, including the non-radial direction distance function method [5], the slacks-based measure (SBM) model with undesirable outputs [7], and the SBM Malmquist–Luenberger model to evaluate the green total factor productivity [7, 22, 23]. By evaluating those green development indexes, researchers have found an imbalance in green development among different regions. Specifically, eastern China has outpaced central and western China in green development [2, 4, 8, 9], and certain cities have higher green development levels or efficiencies than other cities within the same region [6, 7, 18–20, 24].

Researchers have identified a range of impact factors of green development, with research on the effect of green development policy, especially environmental regulations, being particularly notable. The Chinese government has put forward environmental regulations in relation to both command-control and market-related policies, prioritizing sustainable development strategies. Different environmental regulations may affect green development unequally [10, 12, 25]. Variations in policy effectiveness may be one reason the literature offers diverse viewpoints regarding the effects of environmental regulations. For instance, Pan et al. [26] stated that environmental regulations contribute to technological advancement and energy conservation. Jiang et al. [27] found that environmental regulations function negatively for green development, inhibiting enterprises' R&D investment, and thus impeding their green innovation practice. Comparing US versus Swiss hydropower generation regulations, Tonka [28] proposed that defects in the regulatory framework might be the reason for environmental over-exploitation. Mycoo et al. [29] concluded that regulations are ineffective when proper regulatory tools are unavailable: a governance framework should be combined with the available effective market instruments, new technologies, behavioral change, and land tenure reform. Other factors have also been highlighted, including climatic changes and human activities [13], financial activities such as green credit and renewable energy investment [14, 23, 30–32], public participation and regulatory compliance [15], technical progress, and innovation [7, 16, 17, 33, 34].

Since the "economic geography" theory was put forward [35], the spatial correlations of economic activities of agglomeration have drawn attention in many contexts [36, 37], including economic development [38, 39] and financial sectors [40, 41]. Spatial correlations among regions also exist in the context of green development. For instance, green environment efficiency differs across regions and shows solid spatial dependence [42, 43]. This spatial interdependence is prominent in specific indicators such as carbon dioxide emissions, carbon intensity, and water resources [44–46] and in single factors like environmental regulation, technological innovation, and pollution control programs [47–49], as well as the more comprehensive green development indicators of green competitiveness [50] and green total factor productivity [30].

## Methodology

### Overall methodology design

The present study utilized the spatial correlation network distribution characteristics of China's green development. It first developed a system of 55 indicators to evaluate China's provincial green development level, measuring the weight value of each indicator using the entropy weight method and calculating the green development index of each province using the linear weighted sum method. After running the nonlinear Granger causality test, it then applied SNA to construct the spatial association network of China's green development, identifying both whole network and ego network characteristics. Next, it used BMA to analyze the role and status of each province in the spatial correlation network of green development. Provinces were categorized into several subgroups, and the spatial spillover characteristics among those subgroups were further discussed. QAP analysis (including QAP correlation analysis and QAP regression analysis) was then applied to investigate factors influencing the spatial spillovers in the green development network.

### Calculation of green development levels

**Establishment of the green development evaluation system.**   A comprehensive evaluation system of green development was constructed from 55 fundamental indicators

categorized in three dimensions: green economic growth, environmental resource capability, and governmental policy support, as listed in Table 1. In line with the scientific, systematic, practical, and comparability principles and the dimensions identified in previous literature [2, 4, 39], green development level was evaluated on economic (i.e., green economic growth), environmental (i.e., environmental resource capacity), and social (i.e., governmental policy support) dimensions. The economic dimension must fully consider the unification of economic development, green productivity, industrial structure, innovation capability, and economic openness [4, 18]. Therefore, our evaluation system includes 17 indicators such as GDP per capita, energy consumption per unit of GDP, $CO_2$ emissions per unit of GDP, the proportion of transaction value in technical markets in GDP, and the labor productivity of different industries. The environmental dimension should include indicators of environmental greening, pollution prevention, and carbon balance [4, 18]. Accordingly, this study applied 15 indicators such as forest coverage, proportion of nature reserve area to land area, proportion of $CO_2$ emissions to land area, consumption of chemical fertilizers, and pesticide use per unit of cultivated land area. The social dimension consists of 14 indicators, such as proportion of investment in environmental pollution control to GDP, urban green space per capita, and urban sewage treatment rates, covering aspects of industrial pollution control and infrastructure construction [4, 18].

**Data samples, sources, and normalization.** All data were extracted from China Statistical Yearbooks and China's Provincial Statistical Yearbooks from 2006 to 2019. Of the 34 Chinese provincial administrative regions, 30 were selected as research objects because data were missing for four provinces (Taiwan, Tibet, Hong Kong, and Macao).

Because of the differences in the units of each statistical index, the indexes were then standardized. The positive and negative indicators were standardized according to Formula (1) and Formula (2) separately:

$$y_{ij} = \frac{x_{ij} - \min(x_{ij})}{\max(x_{ij}) - \min(x_{ij})} \tag{1}$$

$$y_{ij} = \frac{max(x_{ij}) - x_{ij}}{\max(x_{ij}) - min(x_{ij})} \tag{2}$$

where $y_{ij}$ is the standardized index value, and $x_{ij}$ is the original value of the $j$-th index in the $i$-th province.

**Entropy weight method.** Given that different indicators have different weights in the green development evaluation system, we applied the entropy weight method to decide the importance of each indicator [4, 18, 19]. The green development evaluation system could be established, and China's provincial green development level could be calculated consequently based on the importance of each indicator. The entropy weight method is an objective weighting method that determines the objective weight according to the indicators' variability. Generally, the more significant the degree of variation for an indicator's value, the smaller the information entropy (the higher the information amount), and the more critical the role played by the indicator in the comprehensive evaluation. Compared with subjective weight assignment methods (e.g., the Delphi method), the entropy weight method is more accurate because it avoids the deviation introduced by human factors [51].

We modified the weight of each index based on its entropy value, which we calculated using the fuzzy evaluation matrix and the output information entropy by judging that index's dispersion degree [51]. We then calculated the overall green development level by the linear

**Table 1. Evaluation index system of China's green development.**

| Index | Calculation | Unit | Direction |
|---|---|---|---|
| **Green economic growth** | | | |
| A1: GDP per capita | GDP/Total population | Yuan | + |
| A2: Energy consumption per unit of GDP | Total energy consumption/GDP | Ton by standard coal/ten thousand yuan | − |
| A3: Proportion of non-fossil energy consumption | Non-fossil energy consumption/Total energy consumption | % | + |
| A4: $CO_2$ emissions per unit of GDP | $CO_2$ emissions/GDP | Ton/ten thousand yuan | − |
| A5: $SO_2$ emissions per unit of GDP | $SO_2$ emissions/GDP | Ton/ten thousand yuan | − |
| A6: COD (Chemical Oxygen Demand) emissions per unit of GDP | Discharge amount of COD/GDP | Ton/ten thousand yuan | − |
| A7: NO$x$ emissions per unit of GDP | NO$x$ emissions/GDP | Ton/ten thousand yuan | − |
| A8: Proportion of transaction value in technical markets in GDP | Transaction value in technical markets/GDP | % | + |
| A9: Labor productivity of primary industry | Value-added of primary industry/Employment in primary industry | Yuan per capita | + |
| A10: Land output rate | Gross agricultural output value/Sown area of farm crops | % | + |
| A11: Irrigation saving rate | Water-saving irrigated area/Irrigated area | % | + |
| A12: Proportion of irrigation area | Irrigated area/Area of cultivated land | % | + |
| A13: Labor productivity of secondary industry | Value-added of secondary industry/Employment in secondary industry | Yuan per capita | + |
| A14: Water consumption per unit industrial added value | Industrial water consumption/Industrial added value | Ton/yuan | + |
| A15: Proportion of added value in the tertiary industry | Value-added of tertiary industry/GDP | % | + |
| A16: Proportion of employment in the tertiary industry | Employment of tertiary industry/Total employment | % | + |
| A17: Labor productivity of tertiary industry | Value-added of tertiary industry/Employment in the tertiary industry | % | + |
| **Environmental resource capacity** | | | |
| B1: Water resources per capita | Total water resources/Total population | $m^3$/person | + |
| B2: Forest area per capita | Forest area/Total population | $m^2$/person | + |
| B3: Forest coverage | Forest area/Land area | % | + |
| B4: Proportion of nature reserves area to land area | Nature reserve area/Land area | % | + |
| B5: Wetland coverage | Area of wetland/Land area | % | + |
| B6: Proportion of $CO_2$ emissions to land area | $CO_2$ emissions/Land area | kg/mu* | − |
| B7: $CO_2$ emissions per capita | $CO_2$ emissions/Total population | kg per capita | − |
| B8: $SO_2$ emissions per unit land area | $SO_2$ emissions/ Land area | kg/hectare | − |
| B9: $SO_2$ emissions per capita | $SO_2$ emissions/Total population | kg per capita | − |
| B10: COD emissions per unit land area | COD emissions/Land area | kg/mu | − |
| B11: COD emissions per capita | COD emissions/Total population | kg per capita | − |
| B12: Ammonia nitrogen emissions per unit land area | Ammonia nitrogen emissions/Land area | kg/mu | − |
| B13: Ammonia nitrogen emissions per capita | Ammonia nitrogen emissions/Total population | kg per capita | − |
| B14: Consumption of chemical fertilizers per unit of cultivated land area | Consumption of chemical fertilizers/Area of cultivated land | kg/mu | − |
| B15: Pesticide use per unit of cultivated land area | Pesticide usage/Area of cultivated land | kg/mu | − |
| **Governmental policy support** | | | |
| C1: Proportion of investment in environmental pollution control to GDP | Investment in environmental pollution treatment/GDP | % | + |
| C2: Completed amount of investment in returning grain plots to forests | Forestry investment completed/Area of cultivated land | Yuan/mu | + |
| C3: Urban green space per capita | Green area/Total urban population | % | + |

(*Continued*)

**Table 1.** (Continued）

| Index | Calculation | Unit | Direction |
|---|---|---|---|
| **Green economic growth** | | | |
| C4: Urban sewage treatment rate | Total quantity of urban wastewater treated/Urban wastewater discharged | % | + |
| C5: Hazard-free treatment rate of urban household waste | Total quantity of urban household waste hazard-free treated/ Urban household waste | % | + |
| C6: Number of public transport passengers per 10,000 urban residents | Number of public vehicles in operation (buses, trolleybuses, etc.)/ Total urban population | Number per capita | + |
| C7: Greening coverage rate of built-up area | Green area of urban built-up area/Built-up area | % | + |
| C8: Newly increased afforestation area per capita | Area of afforestation/Total population | $m^2$ per capita | + |
| C9: Urban green park space per capita | Green park areas/Total urban population | $m^2$ per capita | + |
| C10: Proportion of R&D internal expenditure in fixed asset investment | Intramural expenditure on R&D/Investment in fixed assets | % | + |
| C11: Degree of industrial upgrade | Tertiary industry output value/Secondary industry output value | % | + |
| C12: Education expenditure per capita | Expenditure for education/Total population | Yuan | + |
| C13: Number of doctors per capita | Number of doctors/Total population | Number per capita | + |
| C14: Popularity of toilets in rural areas | Number of households using sanitary toilets/Total peasant household | % | + |

Note: *Mu* is the traditional unit for measuring land area in China. One *mu* equals 0.067 *hectares*.

weight sum method using entropy weights. The calculation steps are as follows.

$$p_{ij} = \frac{y_{ij}}{\sum_{i=1}^{m} y_{ij}} \tag{3}$$

where $p_{ij}$ is the proportion of $i$-th evaluation object under the $j$-th index, $m = 1, 2, \ldots, 138$. The information entropy $e_j$ and the difference coefficient $d_j$ were then calculated:

$$e_j = -\frac{1}{\ln m} \sum_{i=1}^{m} p_{ij} \ln p_{ij} \tag{4}$$

$$d_j = 1 - e_j \tag{5}$$

Then, the index weight $w_j$ was obtained:

$$w_j = \frac{d_{ij}}{\sum_{j=1}^{n} d_{ij}} \tag{6}$$

where $n = 1, 2, \ldots, 29$. Finally, the green development level was calculated as follows:

$$U = \sum_{i=1}^{n} y_{ij} w_j \tag{7}$$

## Nonlinear Granger causality test

Before conducting further analysis, it is necessary to identify the relationship between nodes according to the widely used Granger causality test [52–54]. To overcome the insufficiency of the Granger causality test in revealing nonlinear causal relations, this study used the nonlinear test instead of the traditional linear test [55, 56]. The null hypothesis was set as follows: with $y$ being not the nonlinear Granger reasons of $x$ since $y_1, y_2, \ldots, y_t, y_{t+1}$ was independent of $x_1, x_2, \ldots, x_t, x_{t+1}$. y was considered to be a strictly nonlinear Granger cause of $x$ if the null

hypothesis was rejected [56]. The nonlinear Granger causal test results are displayed in S1 Table, a 0–1 matrix representing each province's nonlinear correlated relationship with every other province; a value of 0 denotes the nonexistence of such a relationship, while a value of 1 indicates that the relationship exists [56]. Having passed this test, we were then able to analyze the whole and ego network characteristics of the spatial correlation structures of China's green development utilizing SNA, to identify the core and peripheral regions by BMA, and finally to investigate those impacting factors through the QAP.

## Social network analysis

SNA was first developed by White et al. [57] and then extended to multiple fields such as engineering, sociology, management, computer science, and behavioral research [58–61]. SNA assumes that relationships among social actors are widespread. Thus, according to the graph theory and algebraic model, SNA measures the network characteristics among those social actors [62, 63]. SNA goes beyond almost all other spatial analysis methods (e.g., spatial correlation models) by explaining each node's features through estimations of the whole and ego characteristics of the network [11]. The whole network features mainly analyze the association and structure among members in the network; the ego network mainly analyzes the status and role of each member of the network [64].

**Whole network characteristics analysis.** We selected six indicators to represent the overall characteristics of China's green development network consisting of 30 provinces. The indicators are network density ($D$), network connectivity ($C$), network efficiency ($E$), network hierarchy degree ($H$), average clustering coefficient ($A$), and average path length ($AL$) [57]. They were calculated as follows:

$$D = \frac{L}{N(N-1)} \tag{8}$$

where $L$ is the number of relationships between each province, and $N$ is the number of provinces [57]. The network density reveals the association among the provinces in the network: the higher this indicator, the more relationships exist among those provinces [57].

$$C = 1 - \frac{V}{\frac{N(N-1)}{2}} \tag{9}$$

where $V$ is the number of node pairs in which one node is unreachable from the other [57]. The network connectivity reflects the accessibility of the network: the higher this indicator, the more robust the overall network [57].

$$E = 1 - \frac{M}{max(M)} \tag{10}$$

where $M$ is the number of redundant relationships in the network, and $max(M)$ is the maximum number of those relationships [57]. The network efficiency indicates the existence of redundant relationships: the lower this indicator, the more associated channels across different provinces [57].

$$H = 1 - \frac{K}{max(K)} \tag{11}$$

where $K$ is the number of symmetrically accessible node pairs in the network, and $max(K)$ is the maximum number of those pairs [57]. The network hierarchy reflects the asymmetry in provincial accessibility: the higher this indicator, the more asymmetrical the accessibility

among the provinces [57].

$$A = \frac{1}{N} \sum_{i=1}^{n} \frac{E_i}{R_i(R_i - 1)} \tag{12}$$

where $E_i$ is the number of existing associations between the $i$-th province and its neighbors, and $R_i$ is the number of that province's neighbors [57]. The average clustering coefficient measures the agglomeration degree among the provinces: the higher this indicator, the greater the agglomeration extent [57].

$$AL = \frac{1}{N(N-1)} \sum_i \sum_j d_{ij} \tag{13}$$

where $d_{ij}$ is the shortest path between the $i$-th and the $j$-th provinces [57]. This indicator reflects the associated efficiency of the network, as it measures the shortest average path between all node pairs in the network [57].

**Ego network characteristics analysis.** We chose three indicators to examine single provincial characteristics in the network: point centrality ($PC_i$), closeness centrality ($C_{AP_i}^{-1}$), and betweenness centrality ($BC_i$) [65]. They were calculated as follows:

$$PC_i = \frac{n}{N-1} \tag{14}$$

where $n$ is the number of relationships directly connected between the focal province and the other provinces [65]. The point centrality reveals the position of the focal province in the network: a higher level of this indicator indicates a core role for the focal province in the green development network [65].

$$C_{AP_i}^{-1} = \sum_{j=1}^{n} d_{ij} \tag{15}$$

The closeness centrality reveals the extent to which the focal province is not controlled by the other provinces in the network. A high level of closeness centrality also indicates a core role for that focal province in the green development network [65].

$$BC_i = \frac{\sum_j^N \sum_k^N b_{jk}(i)}{3N^2 - 3N + 2} \tag{16}$$

where $i \neq j \neq k, j < k, b_{jk}(i) = g_{jk}(i)/g_{jk}$, and $g_{jk}$ is the number of shortcuts between the $j$-th and $k$-th provinces [65]. The betweenness centrality reflects a province's ability to control the other provinces in the network. Therefore, as with the other two single characteristic indicators, a high level of betweenness centrality also indicates a core role for the focal province in the green development network [65].

## Block model analysis

The core–periphery theory is an essential analytical tool in economic geography for studying spatial spillover characteristics by identifying the positions of different nodes in the spatial network [66]. Each node's core–periphery position in the spatial network can be identified through its centrality [66]. The core nodes are closely connected and tend to agglomerate subgroups [66]. The periphery nodes are relatively independent (or have few relationships with other nodes), and so they relate to the core nodes instead of agglomerating subgroups [66].

The spatial network's overall core–periphery structure is clarified by analyzing the correlation density between the core and periphery nodes [66].

We used BMA to categorize the provinces in the green development network into several subgroups. BMA is a method of dividing the nodes in a network into discrete subgroups called "blocks" according to specific standards [59]. In doing this, the position of each node, the relationship between blocks, and the network's overall topology structure can be intuitively described [59]. The convergence of iterated correlations method is commonly used to analyze the block model. It measures each node according to specific standards and then establishes a relationship image matrix in which blocks are assigned values of 0/1; thus it incorporates nodes into different blocks [65]. This study used the density criterion α as the measurement standard to establish the image matrix [67].

## Quadratic assignment procedure analysis

We used QAP analysis to investigate the influence factors of the interrelationship in the green development network. QAP is an SNA method for studying the relationship between different relations existing in a network; that is, it examines the correlation and regression between two relationship matrices. This method is based on permutations of matrices. It compares the corresponding lattice values of pairwise matrices, provides the correlation coefficients between them, and performs a non-parametric test on the coefficients [68]. Specifically, the method consisted of three steps. First, the correlation coefficient between two known matrices was calculated: the matrix was converted into a long vector, and the correlation coefficient between them was calculated [69]. Second, the rows and columns of one matrix were randomly replaced. Then the correlation coefficient between the replaced matrix and the other matrix was calculated, and this operation was repeated many times to achieve a distribution [69]. Third, the correlation coefficient calculated in the first step was compared with the distribution achieved in the second step to check for statistically significant differences [69]. This approach allowed the influencing factors' effects on regional spatial correlation to be investigated [11]. QAP analysis includes QAP correlation analysis and QAP regression analysis [69]. The former investigates the correlation between two relationship matrices and between an attribute and a relationship [69]. The latter examines the regression relationship between multiple matrices and the focal matrix [69].

Recall that we constructed China's green development index from 55 indicators covering the three aspects of economics, environment, and society. The regional differences in those three subindices are possibly the impact factor affecting the closeness of spatial correlation; moreover, spatial adjacency and close geographic distance enhance the inter-region correlation [67]. Accordingly, we used green economic growth ($E_C$), environmental resource capability ($R_C$), governmental policy support ($P_C$), spatial adjacency ($W$), and geographic distance ($D$) to represent the influencing factors of the green development network ($G$). We constructed the QAP analysis model as follows:

$$G = F(E_C, R_C, P_C, W, D) \tag{17}$$

## Empirical results

### Results of social network analysis

We examined the spatial network characteristics of China's green development using UCINET software. The total number of relationships was 227. Table 2 presents the whole network characteristics of China's green development. The network density was relatively high ($D = 0.2609$), indicating that a complex network structure represents green development in

**Table 2. Whole network characteristics of China's green development.**

| Characteristic | Network density | Network connectivity | Network efficiency | Network hierarchy degree | Average clustering coefficient | Average path length |
|---|---|---|---|---|---|---|
| Value | 0.2609 | 1 | 0.574 | 0.067 | 0.892 | 1.834 |

China. The network connectivity ($C = 1$) implies that all provinces were in the network. The high network efficiency ($E = 0.574$) indicates redundant relationships in the network or that the network structure was relatively stable, and the nodes were tightly connected with each other. The low network hierarchy degree ($H = 0.067$) shows that the hierarchy in the network had gradually been weakened, and that the provinces with different green development levels were possibly having spillover effects on other provinces. The clustering coefficients and average path length ($A = 0.892$, $AL = 1.834$) imply that China's green development network reflects the prominent small-world characteristics. Altogether, the results show that China's green development has a complex network structure.

Table 3 shows the ego network characteristics of each province. The point centrality ($PC_i$) has a mean value of 46.437, much larger than its standard deviation (9.868). Similarly, the closeness centrality ($C_{AP_i}^{-1}$) has a mean value of 65.382, much larger than its standard deviation (4.243), and the betweenness centrality ($BC_i$) has a mean value of 1.913 and a standard deviation of 1.180. These results indicate that there is little variance among the green development network provinces. Moreover, the centralizations of point centrality, closeness centrality, and betweenness centrality are 17.981, 27.157, and 13.145, respectively, which taken together suggest that the green development network represents a spatially divergent trend. Specifically, the provinces of Hunan, Tianjin, Zhejiang, Henan, and Xinjiang ranked highest in point centrality, whereas Jiangxi, Heilongjiang, Jilin, Shaanxi, and Sichuan ranked the lowest. For closeness centrality, Hunan, Tianjin, Zhejiang, Henan, and Xinjiang ranked highest, and Jiangxi, Heilongjiang, Jilin, Shaanxi, and Sichuan ranked lowest. Similarly, for betweenness centrality, Hunan, Tianjin, Henan, Guangxi, and Yunnan ranked highest, and Jiangxi, Jilin, Shaanxi, Shanghai, and Shandong ranked lowest. The general picture is one of Hunan, Tianjin, Zhejiang, Henan, and Xinjiang occupying core positions in China's green development network, with Jiangxi, Jilin, Shaanxi, Heilongjiang, and Sichuan occupying peripheral positions.

## Results of block model analysis

For the BMA, the current study divides the 30 sample provinces into four blocks using the α-density index evaluation method suggested by Wasserman and Faust [70]. The blocks' α-density matrix and image matrix are shown in Table 4. Block I contains nine provinces: Beijing, Guizhou, Yunnan, Zhejiang, Sichuan, Liaoning, Jiangxi, Hainan, and Ningxia. Block II includes seven provinces: Hunan, Inner Mongolia, Hubei, Fujian, Shanghai, Henan, and Guangxi. Block III consists of eight provinces: Anhui, Hebei, Shanxi, Guangdong, Chongqing, Shandong, Tianjin, Gansu. Block IV consists of six provinces: Heilongjiang, Jiangsu, Shaanxi, Qinghai, Jilin, Xinjiang.

Table 5 displays the spillover relations among the four blocks (intra- and inter-block). There were 227 spatial correlation relationships, 50 generated from provinces within a single block and 177 between different blocks. Thus we can see that the spatial correlation of green development in China occurs mainly between blocks. In terms of gross analysis, Block I sent out the most considerable spillovers, while Block IV spilled out the least. The number of extrovert spillovers received by Block I (65) was twice that received by Block IV (27). Because of the different number of provinces contained in the four blocks, it is impossible to evaluate scientifically the role of each block in the green development network at the aggregate level only.

**Table 3. Ego network characteristics of China's green development.**

| Province | Point centrality | | | | Closeness centrality | | Betweenness centrality | |
|---|---|---|---|---|---|---|---|---|
| | Extrovert degree | Introvert degree | Total degree | Rank | Degree | Rank | Degree | Rank |
| Beijing | 6 | 8 | 41.379 | 18 | 63.043 | 18 | 1.246 | 19 |
| Tianjin | 12 | 10 | 58.621 | 2 | 70.732 | 2 | 4.037 | 2 |
| Hebei | 9 | 6 | 41.379 | 19 | 63.043 | 19 | 1.58 | 16 |
| Shanxi | 9 | 6 | 51.724 | 8 | 67.442 | 8 | 2.184 | 10 |
| Inner Mongolia | 10 | 5 | 48.276 | 12 | 65.909 | 12 | 2.057 | 13 |
| Liaoning | 4 | 8 | 41.379 | 20 | 63.043 | 20 | 1.148 | 21 |
| Jilin | 7 | 2 | 27.586 | 29 | 58.000 | 29 | 0.519 | 29 |
| Heilongjiang | 5 | 4 | 31.034 | 28 | 59.184 | 28 | 0.85 | 25 |
| Shanghai | 6 | 7 | 37.931 | 23 | 61.702 | 23 | 0.776 | 27 |
| Jiangsu | 10 | 6 | 51.724 | 9 | 67.442 | 9 | 2.031 | 14 |
| Zhejiang | 7 | 10 | 58.621 | 3 | 70.732 | 3 | 2.469 | 9 |
| Anhui | 7 | 10 | 48.276 | 13 | 65.909 | 13 | 1.469 | 18 |
| Fujian | 7 | 8 | 48.276 | 14 | 65.909 | 14 | 1.93 | 15 |
| Jiangxi | 4 | 7 | 27.586 | 30 | 58.000 | 30 | 0.452 | 30 |
| Shandong | 11 | 0 | 37.931 | 24 | 61.702 | 24 | 0.82 | 26 |
| Henan | 10 | 10 | 58.621 | 4 | 70.732 | 4 | 3.466 | 3 |
| Hubei | 12 | 6 | 48.276 | 15 | 65.909 | 15 | 2.717 | 6 |
| Hunan | 11 | 14 | 68.966 | 1 | 76.316 | 1 | 5.725 | 1 |
| Guangdong | 8 | 3 | 37.931 | 25 | 61.702 | 25 | 0.857 | 23 |
| Guangxi | 5 | 10 | 51.724 | 10 | 67.442 | 10 | 3.069 | 4 |
| Hainan | 4 | 10 | 48.276 | 16 | 65.909 | 16 | 1.553 | 17 |
| Chongqing | 7 | 11 | 51.724 | 11 | 67.442 | 11 | 2.181 | 11 |
| Sichuan | 3 | 8 | 37.931 | 26 | 61.702 | 26 | 0.851 | 24 |
| Guizhou | 7 | 9 | 55.172 | 6 | 69.048 | 6 | 2.583 | 8 |
| Yunnan | 7 | 12 | 55.172 | 7 | 69.048 | 7 | 3.023 | 5 |
| Shaanxi | 8 | 6 | 37.931 | 27 | 61.702 | 27 | 0.76 | 28 |
| Gansu | 8 | 9 | 41.379 | 21 | 63.043 | 21 | 1.168 | 20 |
| Qinghai | 8 | 4 | 41.379 | 22 | 63.043 | 22 | 1.03 | 22 |
| Ningxia | 5 | 9 | 48.276 | 17 | 65.909 | 17 | 2.131 | 12 |
| Xinjiang | 10 | 9 | 58.621 | 5 | 70.732 | 5 | 2.71 | 7 |
| Max. | 12 | 14 | 68.966 | — | 76.316 | — | 5.725 | — |
| Min. | 3 | 0 | 27.586 | — | 58.000 | — | 0.452 | — |
| Mean | 8 | 8 | 46.437 | — | 65.382 | — | 1.913 | — |
| Standard deviation | 2.473 | 3.059 | 9.868 | — | 4.243 | — | 1.180 | — |
| Centralization | — | — | 17.981 | — | 27.157 | — | 13.145 | — |

**Table 4. Relationships between blocks: Spillover density matrix and image matrix.**

| Block | Density matrix | | | | Image matrix | | | |
|---|---|---|---|---|---|---|---|---|
| | I | II | III | IV | I | II | III | IV |
| I | 0.222 | 0.19 | 0.208 | 0.074 | 0 | 0 | 0 | 0 |
| II | 0.444 | 0.214 | 0.262 | 0.262 | 1 | 0 | 1 | 1 |
| III | 0.222 | 0.375 | 0.375 | 0.25 | 0 | 1 | 1 | 0 |
| IV | 0.389 | 0.429 | 0.125 | 0.133 | 1 | 1 | 0 | 0 |

Notes: When the network density of each block is greater than the overall network density, the related value in the image matrix is 1; otherwise, it is 0.

**Table 5. Relationships between blocks: Gross analysis and intensity analysis.**

| Block | No. of relationships received | | | | Gross analysis | | Intensity analysis | |
|---|---|---|---|---|---|---|---|---|
| | I | II | III | IV | Total extrovert spillover | Total introvert spillover | Extrovert spillover intensity | Introvert spillover intensity |
| I | 16 | 12 | 15 | 4 | 31 | 65 | 0.157 | 0.328 |
| II | 28 | 9 | 13 | 11 | 52 | 51 | 0.310 | 0.304 |
| III | 16 | 21 | 21 | 12 | 49 | 34 | 0.266 | 0.185 |
| IV | 21 | 18 | 6 | 4 | 45 | 27 | 0.3 | 0.180 |

Therefore, we analyzed each block's position in the network using the intensity index (the ratio of actual spillover to maximum spillover). Regarding the density analysis, Block II had the highest extrovert spillover intensity, while Block I had the lowest; for introvert spillover intensity, Block I had the highest, and Block IV had the lowest.

The four blocks of provinces and their spillover relations are summarized in Fig 1. We name each block according to its feature heterogeneity in spillover density and intensity. The definition of each block is in accordance with its feature heterogeneity in extrovert/introvert spillover intensity and green development level. If the extrovert spillover intensity is significantly greater than the introvert spillover intensity, and the green development level of the members of the block is high, the block is a net spillover block; otherwise, it is a net beneficial

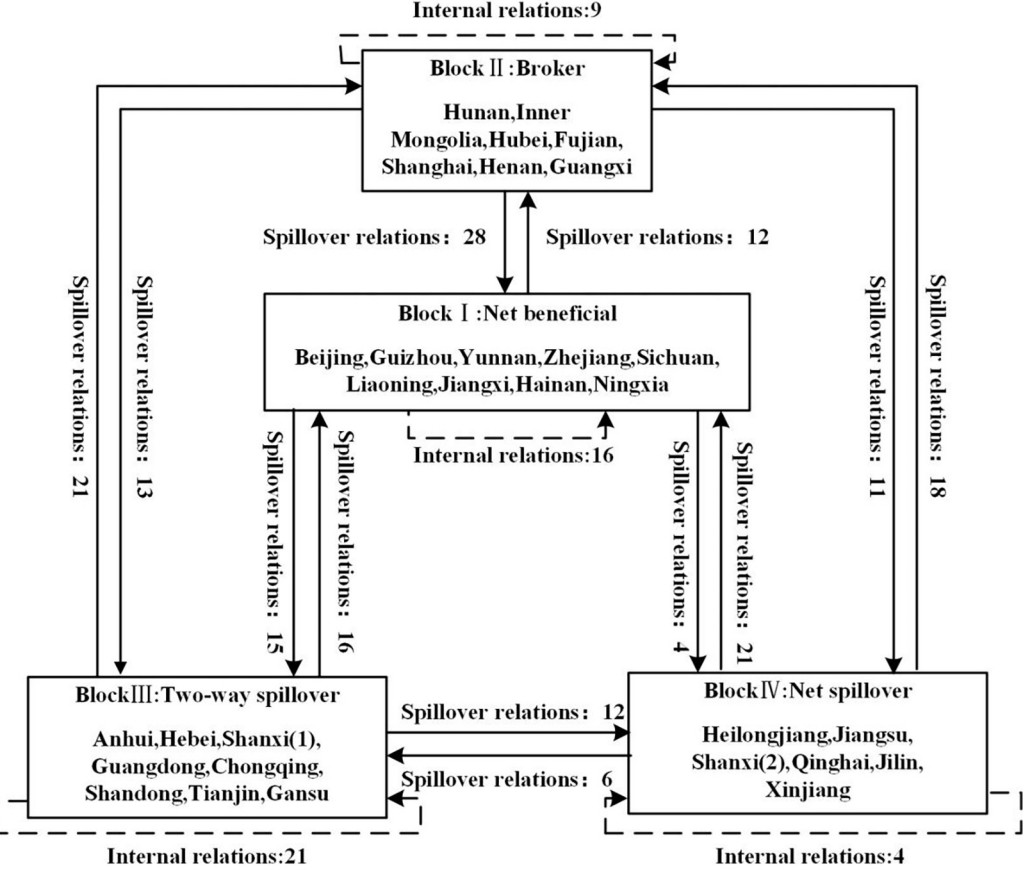

**Fig 1. Four blocks and their spillover relations.**

**Table 6. QAP correlation analysis.**

| Variable | Correlation coefficient | Significance | Correlation coefficient mean | STD | MIN | MAX |
|---|---|---|---|---|---|---|
| $E_C$ | −0.206 | 0.041 | 0.000 | 0.061 | −0.204 | 0.232 |
| $R_C$ | −0.157 | 0.213 | 0.001 | 0.062 | −0.114 | 0.157 |
| $P_C$ | 0.298 | 0.002 | 0.000 | 0.062 | −0.288 | 0.257 |
| $W$ | 0.145 | 0.000 | 0.001 | 0.078 | −0.158 | 0.152 |
| 0–250 km | 0.224 | 0.024 | 0.000 | 0.042 | −0.237 | 0.281 |
| 250–500 km | 0.151 | 0.030 | 0.000 | 0.051 | −0.121 | 0.158 |
| 500–750 km | 0.018 | 0.132 | -0.003 | 0.042 | −0.101 | 0.131 |
| 750–1,000 km | 0.002 | 0.342 | 0.000 | 0.039 | −0.098 | 0.112 |

block. Suppose the extrovert spillover intensity of the block is close to the introvert spillover intensity. In that case, a block with a relatively high green development level is a broker block, and a block with a relatively low green development level is a two-way spillover block.

Block I has the lowest extrovert spillover intensity (15.7%, mainly on Block III) and also the highest introvert spillover intensity (32.8%, mainly from Blocks II and IV), indicating that this block's green development is highly dependent on other provinces; it accepts noticeably more spillover relations from other blocks than it sends out to other blocks. Therefore, we name Block I the net beneficial block.

Block II has the highest extrovert spillover intensity (31.0%, mainly on Blocks I, III, and IV) and the second-highest introvert spillover intensity (30.4%, from Blocks III and IV). That is, this block has the bidirectional spillover feature, in which it accepts spillover relations similar to those it sends out. This status indicates its core role in China's green development network, and we therefore name Block II the broker block.

Block III is named the two-way overflow block because its extrovert spillover intensity (26.6%, mainly on Block II) is similar to its introvert spillover intensity (18.5%, mainly from Blocks I and II). This block receives more spillover relations from other blocks than it sends out.

Block IV is called the net spillover block, as its introvert spillover intensity (18.0%, mainly from Block II) is much lower than its extrovert spillover intensity (30.0%, mainly on Blocks I and II). This implies that this block receives more spillover relations than it sends out.

## Result of quadratic assignment procedure analysis

QAP correlation analysis shows the relationship between a single influencing factor and the green development's overall spatial relationship. As listed in Table 6, despite the regional differences in environmental resource capacity, all other factors were significantly correlated (at the 1% level) to the overall spatial relationship of green development. Specifically, the regional difference in the greening rate of economic growth was negatively correlated to the overall spatial relationship of green development. In contrast, regional differences in government policy and spatial adjacency were positive. In terms of geographic distance, the coefficients of distances less than 500 km were significantly positively correlated to the overall spatial relationship of green development. Generally, government policy support, spatial adjacency, and geographic distance positively affected the overall spatial relationship of green development.

QAP regression analysis shows the impact of different factors in one province on the green development of the other provinces. As listed in Table 7, for the regression model on geographic distance, despite regional differences in environmental resource capacity, all other factors imposed significant effects (at the 5% level) on the green development network.

**Table 7. QAP regression analysis about the distance matrix.**

| Variable | Unstandardized coefficient | Standardized coefficient | Significance |
|---|---|---|---|
| Constant | 0.035 | 0 | |
| $E_C$ | −0.023 | 0.353 | 0.037 |
| $R_C$ | −0.028 | 0.482 | 0.257 |
| $P_C$ | 0.214 | 0.667 | 0.020 |
| 0–250 km | 0.255 | 0.465 | 0.023 |
| 250–500 km | 0.132 | 0.242 | 0.011 |
| 500–750 km | 0.029 | 0.153 | 0.233 |
| 750–1,000 km | 0.014 | 0.096 | 0.139 |

Specifically, the impact of regional differences in green economic growth on regional differences in green development was negative ($\beta$ = −0.023, $p$ = 0.037). For regional differences in government policy support ($\beta$ = 0.214, $p$ = 0.020) and distances shorter than 500 km ($\beta$ = 0.255, $p$ = 0.023 for 0–250 km; $\beta$ = 0.132, $p$ = 0.011 for 250–500 km), all effects were positive. Additionally, coefficients of geographic distance fell as the distance rose, implying that the influence of geographic distance on regional differences in green development decreased when the distance increased. For the regression model on spatial adjacency, Table 8 confirms robustness, as there were no obvious changes in the coefficients of the independent variables.

## Discussions and conclusion

### Key findings

Using provincial data on the green development index system from 2006 to 2019, this study provides an overview of the spatial network characteristics of China's green development and offers a number of interesting findings.

Holistically, the spatial correlation network of China's green development at the provincial level presents a complex yet stable network structure. Almost all provinces are involved in the extrovert and introvert spillovers of green development. Provinces are all tightly related to each other with a weak hierarchy, and so the overall spatial network presents a typical small-world characteristic. Although previous research has mentioned the spatial correlations that exist in China's green development [42, 43], the current study is among the first to elucidate such network characteristics.

In the core–periphery relationship at the provincial level, Hunan, Tianjin, Zhejiang, Henan, and Xinjiang lie at the center of the green development network, playing critical roles in China's overall green development. Meanwhile, Jiangxi, Heilongjiang, Jilin, Shaanxi, and Sichuan are at the network's periphery. These findings echo previous claims of geographical heterogeneity in green development [6, 7, 18–20, 24] at the provincial rather than the city level. However, our findings are considerably different from previous research that suggested a holistic

**Table 8. QAP regression analysis about the adjacency matrix.**

| Variable | Unstandardized coefficient | Standardized coefficient | Significance |
|---|---|---|---|
| Constant | 0.021 | | |
| $E_C$ | −0.054 | −0.447 | 0.000 |
| $R_C$ | −0.031 | −0.482 | 0.134 |
| $P_C$ | 0.267 | 0.809 | 0.017 |
| $W$ | 0.144 | 0.573 | 0.000 |

developing advantage of eastern areas over central or western areas [2, 4, 8, 9]. Specifically, we find that China's green development network can be categorized into four heterogeneous blocks: a net spillover block, a net beneficial block, a broker block, and a two-way spillover block. Each block plays its comparative advantage in this spatially correlated network, and their linkages with each other become accumulatively prominent. The blocks incorporate provinces without prominent geographical adjacency; that is, the core–periphery role of a province in the overall spatial network of green development is considerably more complex than can be captured by a simple taxonomy based on geographical location.

Finally, regional differences in green economic growth, government policy support, geographic distance, and geographic adjacency are significant factors impacting the shaping of China's green development network. Interestingly, the less difference there is in provinces' green economic growth, the more favorable the conditions for forming a spatial correlation network. One explanation is that a similar degree of green economic development benefits the inter-regional transfer of green production factors and promotes the diffusion of energy conservation and environmental protection technologies among regions. The potential capacity of environmental resources has no significant effect, whereas government policy support positively impacts the green development network. Unlike previous research [44–46], the present study finds no significant impact of environmental resource capacity in shaping the green development network. However, consistent with studies like [27], our findings confirm the importance of government policy support in green development. Policies such as environmental regulation may motivate firms to introduce green production technologies from other regions, thereby strengthening inter-regional connections and promoting the formation of the green development network.

## Policy recommendations

We put forward three recommendations for policymakers based on this study's findings. First, policymakers should fully understand the spatial characteristics of green development, seek innovative ideas for realizing green development, and promote the effective transformation of green development policy. The improvement of green practices depends on more than the region's economic development, environmental protection, and energy consumption structure; for instance, the levels of other regions' green development have a role to play. The spatial correlation effect enables inter-regional collaboration in the process of green development. However, the spatial correlation of green development is not as simple as the province–province relationship; instead, there is a complex and multi-directed spillover relationship between multiple provinces. Therefore, it is necessary for both central and local governments to fully realize and leverage the spatial effects of green development. With the development of economic and technological connections among provinces in China, the spatial correlation of green development has become increasingly close. However, even with 267 relationships in the network, the relations between provinces remain weak. Hence, we should pay more attention to strengthening the spatial correlation of China's green development. It is worth reiterating that when SNA is used, the identification of relationships is not limited to geographically adjacent regions but extends to geographically distant regions.

Second, the current study advocates caution and specificity in environmental regulation policy-making. The findings indicate that regional differences in green economic growth and government policy support are the main factors affecting green development's spatial correlation. Underdeveloped areas tend to formulate relatively loose environmental regulation policies to achieve high-speed economic development. Consequently, high-polluting firms tend to transfer from areas with strict environmental regulation to areas with less environmental

regulation, resulting in the aggravation of pollution in the latter areas. This "race to the bottom" presents a challenge for inter-regional cooperation on China's green development. Nevertheless, if a unified environmental regulation policy is set without regard to each region's heterogeneous economic and environmental characteristics, the overall level of green development will be less efficient. Thus, we conclude that the formulation of environmental regulation policies should take into account not only the unity of policies but also the differences between provinces in level of economic development and energy consumption structure. Only then can policy-making avoid the aggravation of pollution or a race to the bottom in relation to environmental regulation.

Third, policymakers should make full use of government control and market mechanisms in promoting green development. On the one hand, policymakers should actively take measures to improve the green development level of undeveloped regions. For instance, governmental regulation should continue to reduce the provincial gaps in economic, technological, and industrial development, thus strengthening inter-provincial green development technology connectivity and promoting the overall improvement of China's green development level. On the other hand, policymakers should make full use of market powers to reduce administrative intervention and use market mechanisms such as competition and financial means to strengthen mutual communication between the core and peripheral areas. They should prioritize support for the latter regions to reduce China's spatial imbalance of green development.

## Limitations and future research directions

This study has a number of limitations that can inform future research. First, we investigated the spatial characteristics of China's green development in terms of green development levels. Future research can focus on the same topic from the green development efficiency perspective (e.g., developing the green total factor productivity index). Second, this study used the objective entropy weight method to integrate the multiple indicators of green development. This method does not take into account the relative importance of indicators or the subjective intentions of decision makers; it is also limited to dealing with indicators with drastic fluctuations. Future studies should seek alternative methods. Third, this study investigated the spatial correlation, network, and spillover characteristics of China's green development using the green development level index integrated by 55 microscopic indicators. The spatial characteristics of each microscopic feature await in-depth investigation in future studies. Fourth, this study used 13 years of data to describe the spatial correlation, network, and spillover characteristics of China's provincial green development levels. The longer the data span, the more accurate the assessment will be, and researchers should in future extend the time span of the data.

## Supporting information

**S1 Table. Result of nonlinear Granger causal test.**
(DOC)

## Acknowledgments

The authors would like to thank the reviewers and the editor, whose suggestions greatly improved the manuscript.

## Author Contributions

**Conceptualization:** Juan Chen.

**Formal analysis:** Jie Huang.

**Methodology:** Jie Huang, Juan Chen.

**Writing – original draft:** Jie Huang, Juan Chen.

**Writing – review & editing:** Juan Chen.

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
