## [Decision Letter · Decision Letter 0]

19 Jul 2022

PONE-D-22-05185Spatial spillover and impacting factors of green development: A study based on China’s provincial dataPLOS ONE

Dear Dr. Chen,

Thank you for submitting your manuscript to PLOS ONE. After careful consideration, we feel that it has merit but does not fully meet PLOS ONE’s publication criteria as it currently stands. Therefore, we invite you to submit a revised version of the manuscript that addresses the points raised during the review process.

Specifically, the revised version should take into account the comments by the referees: the literature review should be expanded, the text should be edited and written more clearly. 

We look forward to receiving your revised manuscript.

Kind regards,

Petre Caraiani

Academic Editor

PLOS ONE

Journal Requirements:

2. Please include a separate caption for each figure in your manuscript.

Reviewers' comments:

Reviewer's Responses to Questions

**Comments to the Author**

1. Is the manuscript technically sound, and do the data support the conclusions?

Reviewer #1: Partly

Reviewer #2: Yes

2. Has the statistical analysis been performed appropriately and rigorously? 

Reviewer #1: No

Reviewer #2: Yes

3. Have the authors made all data underlying the findings in their manuscript fully available?

Reviewer #1: No

Reviewer #2: No

4. Is the manuscript presented in an intelligible fashion and written in standard English?

Reviewer #1: Yes

Reviewer #2: No

5. Review Comments to the Author

Reviewer #1: This paper studies spatial spillover and impacting factors of green development based on China’s provincial data. The subject is interesting, but the overall quality of the paper shall be improved before publication.

Major points:

1.In the past two years, there have been many literatures on green development and space spillovers. It is suggested to add in the literature review.

2.Table 1: The basis for index selection needs to be given.In addition, positive indicators and negative indicators need to be marked.

3.What is the meaning of Figure 1? Is it just showing the complex network structure? If so, it is recommended to delete.

4.Figure 2 needs to be explained. Different line types should be used to indicate the different degrees of overflow and reception.

5.It is suggested to delete the conclusion part and change the discussion part into conclusion.

6.How to define block1, 2, 3 and 4? The description of block2 and 3 is not rigorous.

7.The paper is not clear as a whole. What is the logical relationship between the results of various methods? The result of entropy weight method is not found in this paper? In addition, the role of this result in the whole paper is not described.

8.Whether the network connectivity (c) index is used or not, the corresponding results are not found in the paper.

Reviewer #2: This manuscript studies the spatial spillover effect and impacting factors on green development at the provincial level in China from a social network perspective, this topic is interesting, the analysis is solid. Yet a few concerns need to be addressed before publication:

1. The English expression of the whole manuscript is hard to read, it needs to be revised and refined. It is recommended to invite polishing institutions to provide services.

2. Figure 2 is very vague, it is suggested to impove the image quality.

3. It is suggested to give the limitations of this study and the future research direction, especially the methods, data, influencing factors, etc.

4. The literature review is suggested to supplement and expand.

5. In general, there is not enough explanation or analysis in Empirical Results as they are very important sections for readers’ understanding.

6. An explanation of why the authors select these 55 fundamental indicators should be provided.

7. Section Results of Block Model Analysis is suggested to expand.

8. Section Result of Quadratic Assignment Procedure Analysis is suggested to expand and give more detail, for example, the permutations, the explanation of QAP correlation and QAP regression, etc.

6. PLOS authors have the option to publish the peer review history of their article (what does this mean?). If published, this will include your full peer review and any attached files.

Reviewer #1: No

Reviewer #2: No

---

## [Author Response · Author response to Decision Letter 0]

8 Jan 2023

We greatly appreciate the opportunity to revise the manuscript for Plos One and sincerely appreciate the encouraging comments and directions given by the editors and review team. We would like to thank the editors and the review team for the constructive comments on our paper. With a concerted effort to address all of the valuable suggestions, we have revised and improved the paper by 1) expanding the literature review, 2) improving the writing and having the manuscript proofread by a professional copy editor, 3) adding more speculate explanations for the methodology and the result,s and 4) modifying other problems reviewers have pointed out.

We highlight all the changes in blue text in our revised manuscript. We have provided our detailed responses to explain how editors’ and reviewers’ points have been included in the revision. Please refer to the document of responses to reviewers.

---

## [Decision Letter · Decision Letter 1]

27 Feb 2023

Spatial spillover and impacting factors of green development: A study based on China’s provincial data

PONE-D-22-05185R1

Dear Dr. Chen,

We’re pleased to inform you that your manuscript has been judged scientifically suitable for publication and will be formally accepted for publication once it meets all outstanding technical requirements.

Kind regards,

Petre Caraiani

Academic Editor

PLOS ONE

Additional Editor Comments (optional):

Reviewers' comments:

Reviewer's Responses to Questions

**Comments to the Author**

1. If the authors have adequately addressed your comments raised in a previous round of review and you feel that this manuscript is now acceptable for publication, you may indicate that here to bypass the “Comments to the Author” section, enter your conflict of interest statement in the “Confidential to Editor” section, and submit your "Accept" recommendation.

Reviewer #1: All comments have been addressed

Reviewer #2: All comments have been addressed

2. Is the manuscript technically sound, and do the data support the conclusions?

Reviewer #1: Yes

Reviewer #2: Yes

3. Has the statistical analysis been performed appropriately and rigorously? 

Reviewer #1: Yes

Reviewer #2: Yes

4. Have the authors made all data underlying the findings in their manuscript fully available?

Reviewer #1: Yes

Reviewer #2: Yes

5. Is the manuscript presented in an intelligible fashion and written in standard English?

Reviewer #1: Yes

Reviewer #2: No

6. Review Comments to the Author

Reviewer #1: (No Response)

Reviewer #2: This manuscript studies the spatial spillover effect and impacting factors on green development at the provincial level in China from a social network perspective, this topic is interesting, the analysis is solid.

I think the current version of this paper is "ok" to be accepted.

7. PLOS authors have the option to publish the peer review history of their article (what does this mean?). If published, this will include your full peer review and any attached files.

Reviewer #1: No

Reviewer #2: No

---

## [Editor Report · Acceptance letter]

6 Mar 2023

PONE-D-22-05185R1 

Spatial spillover and impacting factors of green development: A study based on China’s provincial data 

Dear Dr. Chen:

I'm pleased to inform you that your manuscript has been deemed suitable for publication in PLOS ONE. Congratulations! Your manuscript is now with our production department. 

Kind regards, 

on behalf of

Dr. Petre Caraiani 

Academic Editor

PLOS ONE